# Automated Bowel Sound and Motility Analysis with CNN Using a Smartphone

**DOI:** 10.3390/s23010407

**Published:** 2022-12-30

**Authors:** Yuka Kutsumi, Norimasa Kanegawa, Mitsuhiro Zeida, Hitoshi Matsubara, Norihito Murayama

**Affiliations:** Suntory Global Innovation Center Limited, Research Institute, Seika-cho, Soraku-gun, Kyoto 6190284, Japan

**Keywords:** sound analysis, bowel sound, machine learning, neural network, deep learning

## Abstract

Bowel sound (BS) is receiving more attention as an indicator of gut health since it can be acquired non-invasively. Current gut health diagnostic tests require special devices that are limited to hospital settings. This study aimed to develop a prototype smartphone application that can record BS using built-in microphones and automatically analyze the sounds. Using smartphones, we collected BSs from 100 participants (age 37.6 ± 9.7). During screening and annotation, we obtained 5929 BS segments. Based on the annotated recordings, we developed and compared two BS recognition models: CNN and LSTM. Our CNN model could detect BSs with an accuracy of 88.9% andan F measure of 72.3% using cross evaluation, thus displaying better performance than the LSTM model (82.4% accuracy and 65.8% F measure using cross validation). Furthermore, the BS to sound interval, which indicates a bowel motility, predicted by the CNN model correlated to over 98% with manual labels. Using built-in smartphone microphones, we constructed a CNN model that can recognize BSs with moderate accuracy, thus providing a putative non-invasive tool for conveniently determining gut health and demonstrating the potential of automated BS research.

## 1. Introduction

Digestive tract issues, including inflammatory bowel disease (IBS), have been reported to affect an increasing number of patients worldwide [1]. Although exercise and functional foods, such as probiotics and prebiotics, are often used as countermeasures, it can be difficult for patients to determine the cause of their bowel conditions, and therefore, which treatments would be beneficial. A tool that senses gut health would enable personalized patient care; however, existing diagnostic tools, such as gut microbiota analysis services, X-ray imaging, and endoscopy, are highly invasive, costly, and limited to a hospital environment. Since gut health plays an important role in overall health and well-being, this issue must be addressed urgently.

A simple and recognizable means of sensing gut health is through bowel sounds (BSs), which are used as an indicator of migrating motor complex [2] and can be obtained through auscultation and from electronically recorded voice data. Various BS parameters are known to indicate diseases and other conditions. For example, the loss of BSs can be used to diagnose ileus, and monitoring and quantifying BSs after surgery for gastric cancer can prevent postoperative ileus [3]. In addition, differences in the sound-to-sound interval (SSI) have been reported between healthy subjects and patients with IBS or Crohn’s disease, indicating that BSs can be used to diagnose bowel-related diseases and bowel motility [4,5]. With regard to food, BSs are also expected to be effectively used for food-intake related sensing, such as capturing changes in BSs with coffee and carbonated water intake [6] and changes in BS indices with the amount of energy contained in food [7].

BSs can be analyzed directly by a physician or using a learning model that recognizes sounds collected using a special recording instrument [4,6,8,9,10,11]. For instance, Craine et al. [4] used a method involving stationary-nonstationary filters with wavelet transformation, whereas Wang et al. [7] used an artificial neural network and Sakata and Suzuki [11] used a classification algorithm capturing sound features. Although BS measurement is not invasive and various analysis methods have been established, it requires specialized devices such as stethoscopes and contact-type recording devices that are only available in a hospital environment. This has limited the collection of BS data from healthy people, leading to a lack of basic information on BSs and their therapeutic applications [12]. Smartphones with recording capabilities are ubiquitous worldwide and have been used to acquire and analyze biological sounds. For instance, smartphones have been used to diagnose sleep apnea syndrome by measuring breath sounds and snoring during sleep [13]. The application of machine learning (ML) models has led to advancements in the use of biological sound as a source for medical diagnosis [14]. Recent examples include the detection of COVID 19 detection [15], and chronic heart failure [16]. However, because the accuracy of the measurement has an impact on medical diagnosis, the application of the models in a clinical setting has been challenging.

However, smartphones have not yet been used to measure and analyze BSs. In this study, we constructed a BS recognition model based on sounds obtained by subjects themselves using a smartphone and examined whether the model had similar accuracy to previous models based on BSs obtained using dedicated devices. Since general-purpose recording devices can result in strong noise, we used a convolutional neural network (CNN) that can automatically extract features from input data with high accuracy and low noise, which has previously been used for image, speech, and more recently, BS recognition [8,9]. The CNN model constructed using built-in smartphone microphones could recognize BS with moderate accuracy. Consequently, we developed a simple, less invasive method that allows anyone to easily measure BSs with minimal mental, physical, and economic burden. This method could improve public health by accelerating BS research and increasing our knowledge of gastrointestinal diseases. Furthermore, this study demonstrates the potential of collecting more BS data, which can advance the automated BS research.

## 2. Materials and Methods

### 2.1. Data Collection

This study was approved by the Institutional Review Board of Suntory Holdings (Kyoto, Japan). Study participants were recruited from employees of the Suntory World Research Center (Kyoto, Japan). BS recordings were collected from 100 participants, including 43 men and 57 women between the ages of 20 and 65. All participants provided informed consent.

Apple iPhone 7 (Apple Inc. 2017, Cupertino, CA, USA) smartphones were used for all experiments. A range of iPhone models (6s, 7, 8, SE2, and X) were assessed prior to the study to determine whether the models had different levels of electromagnetic interference. No significant differences in sound quality were observed among the models tested. All study participants were given iPhones with instructions for installing the BS recording app. Since fetal sounds may contaminate BS recordings, we excluded pregnant individuals and those intending to become pregnant during the study.

The test period consisted of four days of answering daily surveys and one day of BS recording. All participants agreed to completely abstain from eating food and drinking alcohol from 10:00 p.m. until the end of the BS recording the following morning. Each participant joined an online session at 9:00 a.m. and was instructed to adopt a seated position (Figure 1a) to measure their own BSs in a location where the noise level was less than 50 dB. This threshold was determined through prior experiments that tested the audible range of the iPhone7 microphone using white noise and the Voice Memo app (Apple Inc. 2016). The iPhone 7 was capable of recording audio signal from 40 Hz–16 kHz, and BSs were audible if the background sound was less than 60 dB. Each participant was asked to place the bottom microphone of the smartphone perpendicular to the abdomen, approximately 9 cm away from the navel in both directions (Figure 1b), and 60 s recordings were made on both sides of the abdomen in the morning and before bedtime, for a total of eight recordings.

### 2.2. Data Analysis

BS recordings were downloaded from a server contracted by Suntory Holdings, Ltd., transformed into a spectrogram, and divided into multiple successive non-overlapping 100 ms segments due to the 48 kHz sampling rate and the 4800 non-uniform fast Fourier transformation (NFFT).

Three human raters trained in BS identification by a medical doctor performed double blinded annotations on all segments using Audacity(R), version 2.4.2, open-source sound editing software [17]. Annotations consisted of binary labeling based on the presence or absence of BSs. Sound data labeled identically by more than two raters was used as the training data set. If adjacent BS labels were more than 10 ms apart, they were treated as independent labels. If adjacent BS labels were within 10 ms, they were treated as one unified label.

### 2.3. Non-BS Characterization

Non-BSs were categorized as follows: (1) circuit noise originating from the smartphone; (2) biological noises originating from participant internal organs, such as heartbeats; (3) background noise originating from the surrounding environment. Only sound files in which the human raters could clearly identify BSs were used for further analysis.

### 2.4. CNN Model Design

ML models are effective for identifying BSs [7,12,14,15,16,18], among which CNNs have been reported to offer better BS recognition accuracy [8]. Another neural network model that has a good reputation for sound recognition is long short-term memory (LSTM), which can process not only single data points, but the entire data sequence [19,20]. Due to the seemingly random nature and dynamic range of BSs, we developed and compared CNN and LSTM-based BS recognition models.

The CNN model consisted of seven components: spectrogram formation (regular convolution), batch normalization and dropout, rectified linear unit activation, max pooling, and fully connected layers (Figure 2). Hyper-parameter experiments determined a frame length of 100 ms with 75% overlap allowed. The total length of the recording was 7740 s. For the input architecture, we used a mini batch size of 32, data height of 3, and data width of 11. One hundred thousand iterations were used to train the model. Dropout was set after the third layer.

First, the 60 s audio file with a sampling frequency of 48 kHz was converted into a WAV file using ffmpeg software (version 3.4.11) [21] and then converted into a mel scale spectrogram using the librosa package (version 0.8.0) in Python. The hanning window function was used to create the time axis component of the spectrogram using the Fourier transform process. A mel spectrogram, wherein the frequencies are converted to the mel scale, was used to display BS findings. Mel scale is ideal for detecting differences in lower frequencies than in higher frequencies. The frequencies of BS are reported to be in the range of 100–1000 Hz [12,20]; therefore, mel spectrogram was chosen for our study.

### 2.5. CNN Model Setup 

The frame length was set to 100 ms with a 75% overlap allowed. It was determined that three building blocks, consisting of two convolution layers and a max pooling layer, were the most optimal for the model (Appendix A). For the number of channels, an 8, 16, 16 combination was determined to be optimal along with a 3 × 3 kernel size (Appendix A).

### 2.6. Model Evaluation

To accurately evaluate model performance, we selected a test set representing the average proportion of BSs and noise in all recordings. To determine how accurately the model could detect BSs over noise, holdout and cross evaluation methods were used. The dataset was divided into nine folds, each containing BS data for 15 subjects with no overlap. For holdout evaluation, a single train-test split was run, meaning that only the first split was tested (‘Split1’ in Figure 3). This test was repeated five times. For cross validation, which runs multiple train-test splits, nine fold cross validations were performed with eight splits and uncorrected labels, and the average results were calculated.

### 2.7. SSI Correlation Analysis

To determine how accurately the CNN model could recognize BSs, we used the correct labels and prediction results to calculate the fasting SSI (Figure 4), which is the interval between BSs defined as the period between the end of one BS and the start of the next BS.

### 2.8. Software 

Models were run using Python script version 3.8. Audio files were processed using librosa package (version 0.8.0), and pytorch (version 1.3.1) was used for network creation and model training. Hyper-parameter tuning was performed using Amazon Ec2 instance (g3xlarge 30.5 GB RAM with 8 GPU, Amazon.com Inc., Seattle, WA, USA) with a Linux operating system.

### 2.9. Statistical Analysis

To compare the performance of our two ML models, four metrics were employed to evaluate accuracy, precision, sensitivity, and the F measure, which provides the harmonic mean of precision and sensitivity, and is a better measure than accuracy when working with an imbalanced binary classification dataset. Pearson’s method was used to test the correlation between manually labeled and predicted BS, with *p* < 0.05 defined as significant.
Accuracy = (TP + TN)/(TP + TN + FP + FN)(1)
Precision = TP/(TP + FP)(2)
Sensitivity = TP/(TP + FN)(3)
F measure = 2∗(Precision∗Sensitivity)/(Precision + Sensitivity)(4)

TP and TN refer to correctly classified BS and non-BS counts, respectively. FP and FN refer to falsely recognized and falsely rejected BS counts, respectively.

## 3. Results

In this study, audio files were collected from 100 individuals. Details of participant age, height, weight, and body mass index (BMI) are included in Table 1 below. The study participants had a wide age range and a variety of body types. All the participants completed the experiments. However, some audio files were not audible by the human raters due to the noise contaminations (Figure 5), so we used 144 audio files from the smartphone for this study.

### 3.1. BSs Can Be Recorded Using a Smartphone Microphone 

With the developed procedure, study participants were able to record BSs using a smartphone microphone (Figure 6). Smartphone microphones generally support a frequency range of 20–20,000 Hz. The mel spectrogram produced in this study showed various types of BSs, indicating that BSs occur within the audible range of smartphone built-in microphones.

### 3.2. Experimental Results 

#### 3.2.1. Holdout Evaluation Results

Our CNN model had an accuracy of 83.9%, precision of 75.7%, sensitivity of 78.6%, and F measure of 77.0%. The LSTM model reached an accuracy of 77.2%, precision of 53.7%, sensitivity of 87.9%, and F measure of 65.8% (Table 2, Figure 7). Thus, the CNN model had better accuracy and F measures than the LSTM model.

#### 3.2.2. Cross Validation Results

Cross validation revealed that the CNN model had an accuracy of 88.9%, precision of 70.5%, sensitivity of 74.9%, and F measure of 72.3%. Meanwhile, the LSTM model reached an accuracy of 82.4%, precision of 53.7%, sensitivity of 87.9%, and F measure of 65.8%. Together, these results indicate that the CNN model had a better accuracy and F measure than the LSTM model (Table 3, Figure 7).

#### 3.2.3. SSI Classification and Correlation Results

To compare the ability of the model for BS recognition against human hearings, the correlation coefficients of the SSI and the sound duration (SD) were calculated (Table 4, Figure 7).

As for the SSI, the holdout evaluation result of the CNN model showed 99.2% correlation between the manual labeling and the prediction, and the cross-evaluation result showed 94.0% of the correlation. The LSTM model results of the holdout and the cross evaluation showed 92.1% and 87.2%, respectively. As for the SD, which is the sum of the total BS sound in a single file, the CNN holdout result showed 96.0% correlation, and the cross evaluation was 90.5%. The LSTM holdout result showed 82.2% correlation, and the cross evaluation was 74.2% (Table 4, Figure 7). It was found that both SSI and SD had the best correlation coefficient with CNN holdout evaluation method.

## 4. Discussion

Gut health is closely related to mental and physical quality of life; however, current diagnostic tests are highly invasive, costly, and limited to hospital settings. Since BSs can be used as a physiological indicator of gut health, we developed a prototype BS recording application using built-in smartphone microphones. Using the smartphone application, 100 subjects recorded BSs at home with a background noise level below 50 dB. Although BS measurement was carried out remotely, live instruction was provided to ensure that all participants placed the smartphone in the correct position for exactly 60 s. Importantly, BSs were audible using the built-in microphone of the smartphone.

Based on the recording data obtained from these experiments, two ML-based models were built. The CNN model, which was tuned for the sound quality of the smartphone microphone, had a higher F measure (77.0%) and moderate accuracy (83.9%), suggesting that the sound input from the built-in microphone is adequate for recognizing BSs using the proposed CNN model. Cross evaluation further confirmed that the CNN model displayed better accuracy and F measures than the LSTM model. Therefore, we calculated the SSI using the CNN model results and compared the correct labels and prediction results to determine whether the model output made biological sense.

In this study, we found that it is possible to acquire BS data using a smartphone at a sensory evaluation level, as confirmed by a gastrointestinal surgeon. The frequency range of the acquired BSs was approximately 100–1000 Hz (Figure 6), consistent with the frequency range of BS acquired using stethoscopes and other dedicated devices [12]. To confirm the audible range of the iPhone 7, we checked the frequency range that can be acquired when white noise is played. The frequency range was found to be 10–10,000 Hz, validating our finding that the iPhone 7 can record BSs. In addition, the smartphone microphone is slightly recessed, meaning that noise is reduced when the microphone is placed close to the body, allowing BSs to be recorded accurately. Although previous studies have used specific devices to detect BSs [22], this study is the first to use smartphones. Notably, we found that the BS recordings made by smartphones were comparable to past studies in terms of accuracy and F measure (Table 5).

Since BSs account for a small percentage of the total measurement time (i.e., low frequency), we created the CNN model to avoid false negatives so as not to miss BSs, with an emphasis on the F measure in addition to accuracy. The SSI obtained from the manual labeling of subjects in this study was 2.00 ± 7.92, which is similar to the results of a previous study conducted by Craine et al. [4]. The correlation between the SSI calculated from manually labeled BSs and that calculated from automatically extracted BSs was very high (R = 0.99, *p* < 0.05), indicating that the model developed in this study can calculate the SSI. Previous studies have reported that the SSI can be used to predict IBS and Crohn’s disease, and that the SSI can change before and after the consumption of beverages (e.g., carbonated water [6]). Thus, our technology could be used to diagnose these diseases and make suggestions to improve quality of life using smartphones.

Although we were able to construct a model to recognize BSs, it remains unclear what meaning can be assigned to different types of BSs. Future cross-sectional studies should therefore clarify the relationship between BSs and other clinical findings (e.g., bowel diseases, stool frequency and shape, and timing of acquisition). Recent studies have shown that the diversity of gut microbiota correlates with gastrointestinal motility [24], and that it may be possible to predict defecation from BSs [25]. Thus, we believe it is important to clarify the relationships between the SSI and other BSs. Since smartphones are not dedicated measuring devices, they detect a lot of background noise from the external environment, especially in the low-frequency band. This could bury important features by decreasing the signal-to-noise ratio. Consequently, it is necessary to consider how and where to apply the device and to keep the device away from nearby electronic devices.

During general auscultation, BS intervals of 2–15 s/min are considered normal, with a decrease in this interval indicating an increase in peristalsis and an increase in this interval indicating a decrease or absence of peristalsis. The mean value of the BSs obtained in this study was small, which is enough to be diagnosed as increased peristalsis during general auscultation. This labeling was conducted using fragments (0.1 s) so small that it is difficult for a person to recognize them without spectrogram visualization; therefore, to establish the extent to which the calculated SSI is related to peristalsis, BS measurements by manometry and colonic transit tests should be performed in parallel, as seen in a previous study [26]. This formula would allow clinical-level gut diagnosis to be performed at home using a smartphone.

## 5. Conclusions

Using a built-in smartphone microphone, we built a CNN model that can recognize BSs with moderate accuracy, thus providing a foundation for new non-invasive tools to conveniently measure gut health. This study has significant implications because it demonstrates the reliability of the CNN model for recognizing BSs using the microphone of a smartphone. Since many people own smartphones, more BS data can be collected; however, the prototype recording app needs to be improved in order to attract more users. In conclusion, we believe this smartphone application will bring gut health care to the next stage by offering a simple BS measurement tool to users, thus improving a healthy lifestyle.

## Figures and Tables

**Figure 1 sensors-23-00407-f001:**
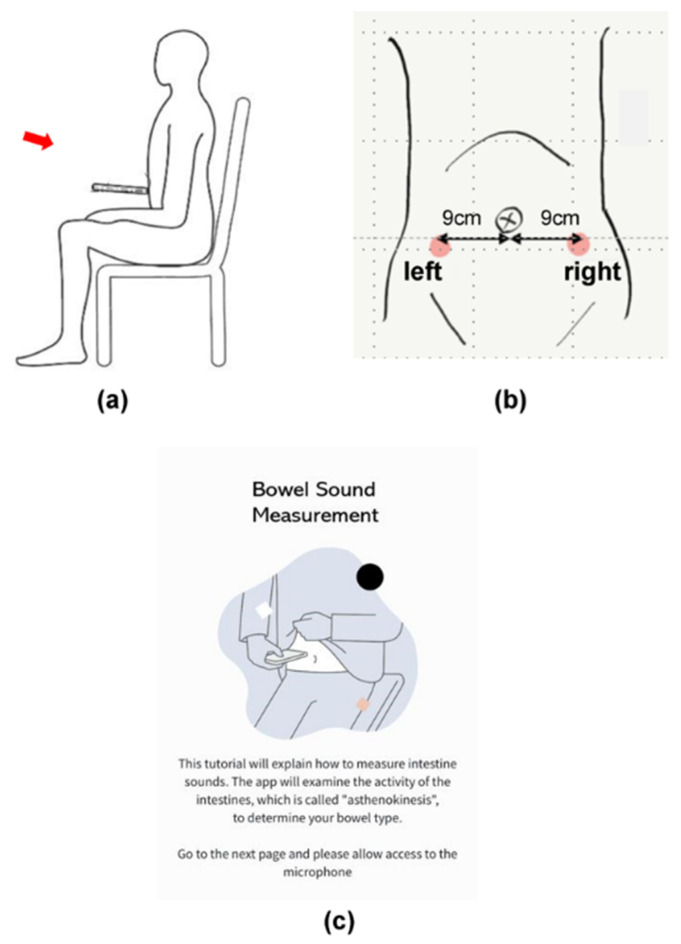
Bowel sound measurement positions. (**a**) Seated recording posture; (**b**) microphone placement over the lower right and left parts of the abdomen; (**c**) a smartphone screen showing the measurement instructions.

**Figure 2 sensors-23-00407-f002:**
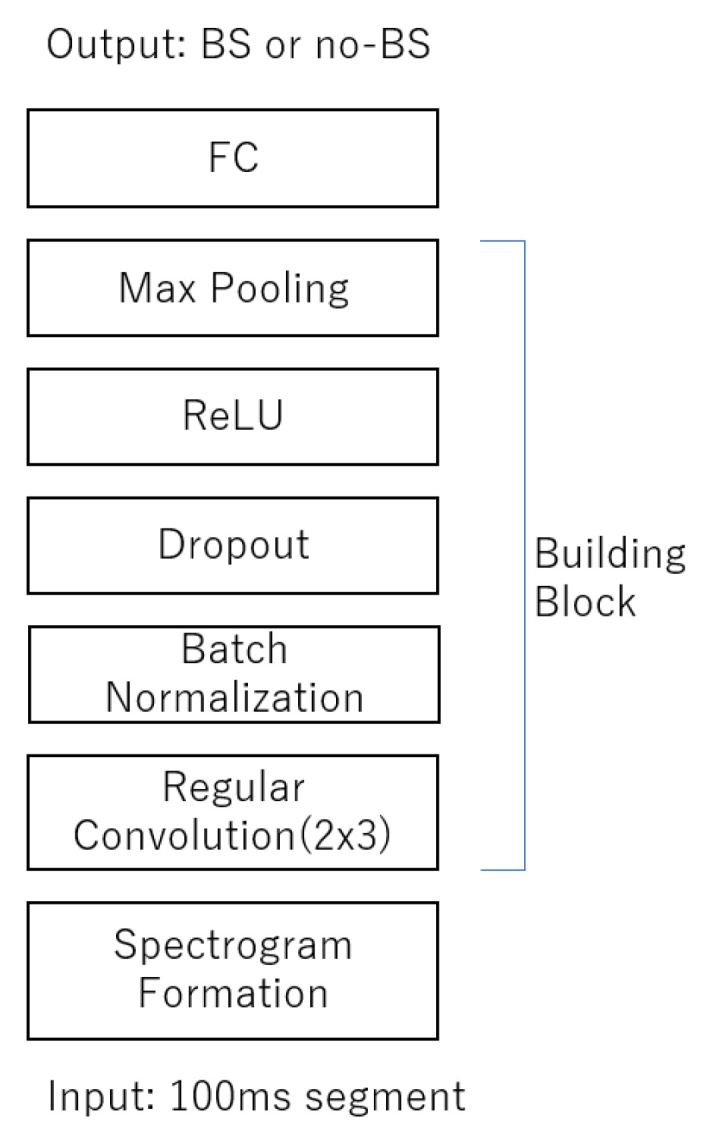
The 100 ms segment of spectrogram was fed to CNN building blocks where it consisted of two sets of three convolution layers, batch normalization, dropout, rectified linear unit (ReLU) activation function, max-pooling layer, and fully connected layer (FC).

**Figure 3 sensors-23-00407-f003:**
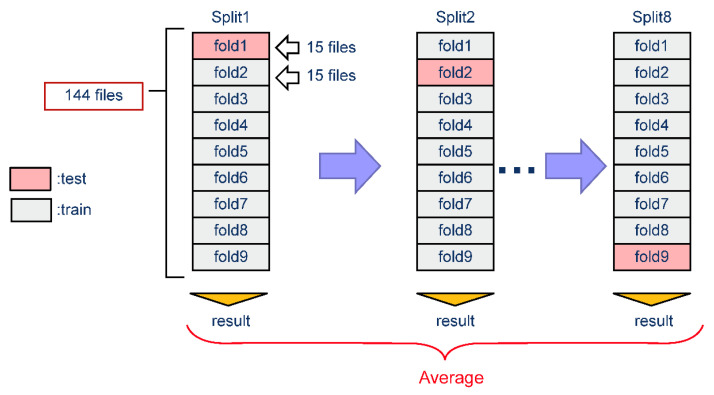
Cross evaluation schematics.

**Figure 4 sensors-23-00407-f004:**
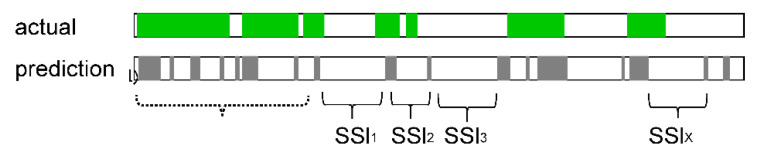
Sound to sound interval was calculated by the length between the end of the bowel sound and the start of the consecutive bowel sound. The green colored area in the top horizontal bar labeled “actual” shows the bowel sound length that have been manually labeled by human raters. The bottom one with gray colored area labeled “prediction” shows the bowel sound length that have been predicted by the models. The interval of the bowel sounds is used as an index called SSI.

**Figure 5 sensors-23-00407-f005:**
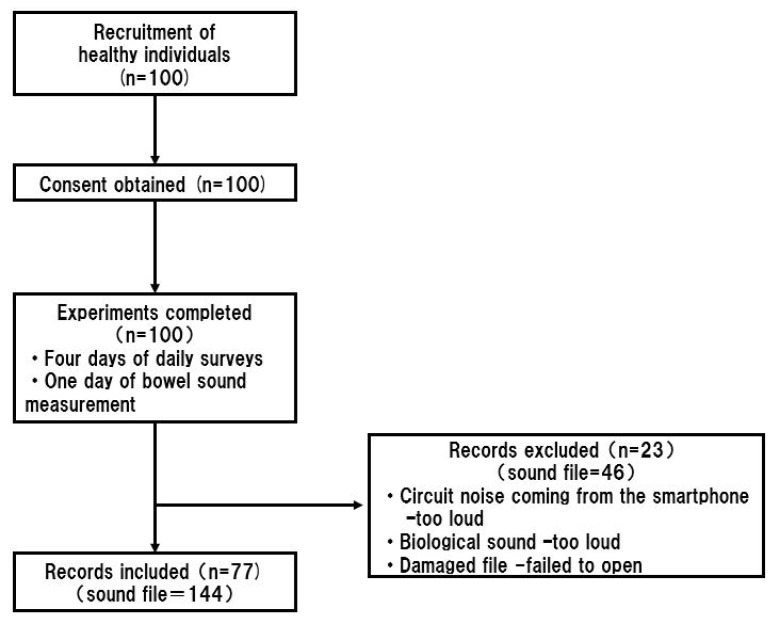
Flowchart of participant recruitment and data acquisition. All 100 participants completed the study. Due to noise contamination, the data for only 77 participants was used to build the machine learning model.

**Figure 6 sensors-23-00407-f006:**
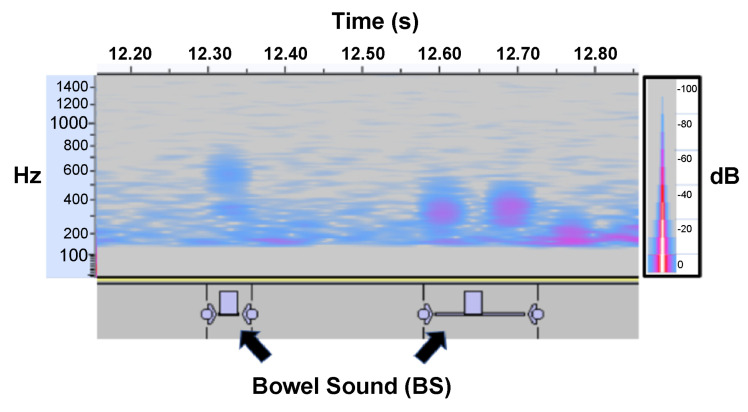
Spectrogram depicting bowel sounds recorded using the bottom microphone of a smartphone held in the right lower quadrant of the abdomen. Individual short BSs with multiple peaks are visible around the 100–800 Hz range.

**Figure 7 sensors-23-00407-f007:**
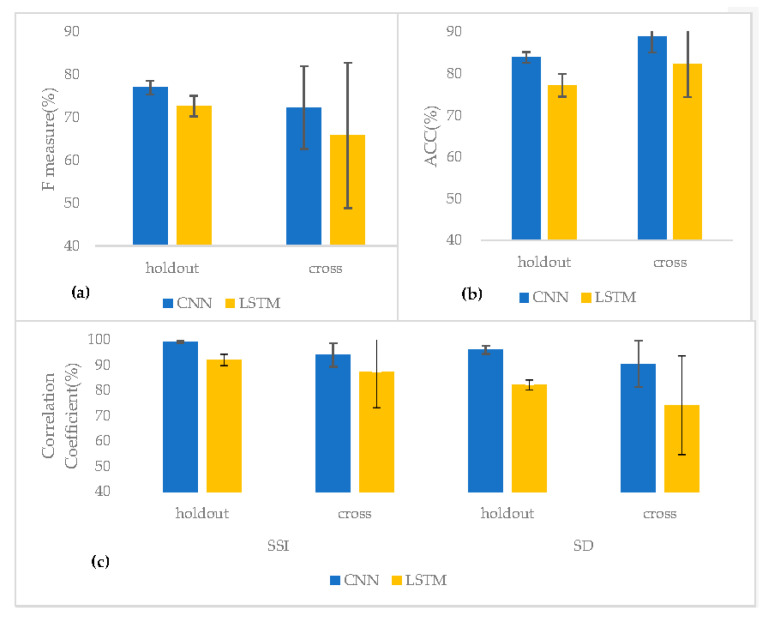
Two BS recognition models; CNN and LSTM are compared by using two evaluation methods; a holdout and a cross evaluation. (**a**) F measure results. (**b**) ACC results. (**c**) The correlation coefficient (%) for sound-to-sound interval (SSI) and sound duration (SD).

**Table 1 sensors-23-00407-t001:** Study participant characteristics.

	All (*n* = 100)	Male (*n* = 43)	Female (*n* = 57)
Age	37.6 ± 9.7	38 ± 9.7	37.3 ± 9.7
Height (cm)	165.7 ± 8.2	172.5 ± 5.2	160.6 ± 6.0
Weight (kg)	58.3 ± 9.6	65.9 ± 7.8	52.6 ± 6.4
Body Mass Index	21.1 ± 2.2	22.1 ± 2.0	20.4 ± 2.1

**Table 2 sensors-23-00407-t002:** Holdout evaluation results for the two models.

	Accuracy	Precision	Sensitivity	F Measure
CNN	0.839 ± 0.013	0.757 ± 0.062	0.786 ± 0.076	0.770 ± 0.016
LSTM	0.772 ± 0.027	0.616 ± 0.035	0.887 ± 0.056	0.727 ± 0.024

**Table 3 sensors-23-00407-t003:** Cross evaluation results for the two models.

	Accuracy	Precision	Sensitivity	F Measure
CNN	0.889 ± 0.038	0.705 ± 0.140	0.749 ± 0.142	0.723 ± 0.097
LSTM	0.824 ± 0.080	0.537 ± 0.216	0.879 ± 0.067	0.658 ± 0.170

**Table 4 sensors-23-00407-t004:** The correlation coefficient of the sound-to-sound interval (SSI) and the sound duration (SD) between manual labeling and prediction.

	SSI	SD
CNN-holdout	0.992 ± 0.004	0.960 ± 0.016
CNN-cross	0.940 ± 0.047	0.905 ± 0.091
LSTM-holdout	0.921 ± 0.022	0.822 ± 0.020
LSTM-cross	0.872 ± 0.140	0.742 ± 0.195

**Table 5 sensors-23-00407-t005:** Summary of recent bowel sound algorithms. This summary is based largely on the results in Table 10 of Ficek et al. [9].

Author	Algorithm	Input Source	ACC	F Measure
Sato et al. [6]	ANN	non-contact microphone	90%	15%
Kumar et al. [23]	SVM	contact microphone	75%	N/A
Liu et al. [19]	CNN	custom design contact microphone	92%	N/A
Ficek et al. [9]	CNN	custom design contact microphone	97%	66% *
Ours	CNN	smartphone microphone	89% **	72%

* F measure for Ficek et al. was calculated by reproducing their models and using the same test dataset as ours (Appendix A). ** Our ACC by a holdout evaluation was 83%, and the ACC by a cross-evaluation was 89%. Since the ACC reported by Ficek et al. is a cross-evaluation result, our cross-validation result instead of the holdout evaluation result was inserted.

## Data Availability

Due to the nature of this research, data is not available at this point.

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
