# Peer review of "Automated Bowel Sound and Motility Analysis with CNN Using a Smartphone"

_sensors, 2022, doi:10.3390/s23010407_

Round 1
Reviewer 1 Report
1. The contributions of the research work should be mentioned in the introduction section.
2. The results in Table 5 require more discussion. How the proposed system is better as compared to existing ones as ACC is 83% in proposed work while it was high in existing work.
3. Results need to discussed in detail.
Authors have developed the CNN based model Bowel sound recognition that was able to achieve moderate accuracy.
1. The background is provided but it needs to be more specific particularly the motivation behind the work and contributions of the proposed work.
2. The results in Table 5 need more elaboration as the results show less ACC than existing ones. Thus, how the proposed work is contributing?
3. References are appropriate and relevant to the study.
4. Quality of Figure 6 can be improved.
Author Response
Dear reviewer,
Thank you so much for taking your time to review my manuscript. Please see the attached file for my answers.
Sincerely,
Yuka

Reviewer 2 Report
See attached file.

Author Response
Dear Reviewer,
Thank you so much for taking your time to review my manuscript. I really appreciate your time and advices you have given to me. Please see the attached document for my answers. I apologize in advance that I still need to work on the references and one figure, but I would like to know if I am on the right path for interpreting some of your comments.
Sincerely,
Yuka

Round 2
Reviewer 2 Report
All of my comments were properly addressed. I suggest the Authors to include the Figure on which they are working on, and to share data in the "Data Availability Statement".
Author Response
Dear Reviewer,
Thank you so much for the comment. I have added a figure in the manuscript (Fig. 7) .
As for the data availability, I discussed with other collaborators but due the nature of the research, participants of this study did not agree for their data to be shared publicly at this moment, so the BS data can not be available.